# Neurofeedback in ADHD: A qualitative study of strategy use in slow cortical potential training

John Hasslinger[1]*, Manoela D'Agostini Souto[1], Lisa Folkesson Hellstadius[1], Sven Bölte[1,2]

1 Center of Neurodevelopmental Disorders (KIND), Centre for Psychiatry Research, Department of Women's and Children's Health, Karolinska Institutet & Child and Adolescent Psychiatry, Stockholm Health Care Services, Region Stockholm, Stockholm, Sweden, 2 Curtin Autism Research Group, School of Occupational Therapy, Social Work and Speech Pathology, Curtin University, Perth, Western Australia

* john.hasslinger@ki.se

**Data Availability Statement:** All relevant data are within the manuscript and its Supporting Information files. The raw interviews and transcripts may contain sensitive information and

## Abstract

Neurofeedback (NF) as a treatment for children and adolescents with attention deficit hyperactivity disorder (ADHD) has gained growing interest in recent years. Most research has been quantitative, focusing on treatment outcomes, while qualitative approaches exploring the treatment process and participants' experiences are scarce. The objective of this study was to examine NF participants' use of cognitive and other strategies for approaching and solving NF tasks, their development over the course of the training and the influence of participants' compliance.

### Method

We collected 130 short semi-structured interviews following treatment sessions from 30 participants with ADHD receiving NF using slow cortical potential training (SCP). The interviews were transcribed verbatim and analyzed using thematic analysis. Themes where evaluated for changes over-time and between participants with high/low treatment compliance. Interviews from 14 participants who had undergone at least five completed interviews were examined in more depths, aiming to establish typical strategy/training profiles.

### Results

We identified 16 strategies covering four domains: cognitive, physiological, emotional and unspecified. Typical of most strategies were that they served as a vehicle to regulate mental arousal. Overall, no clear patterns of changes over time were found. Highly compliant participants reported to use the strategies from the *emotional* domain and the strategy *focus* more frequently than neutral compliant participants did, while neutral compliant participants reported the use of the strategies *muscular activity* and *passivity* more often than participants did with high compliance. Across participants, three strategy profiles were derived, those who handled the task by manipulating their *state of mind* in relation to the NF task, those who were mainly *manifest and concrete* towards the task, and those who were mostly *unaware* of what they were doing. These profiles differed in self-regulatory performance,

cannot be shared publicly, due to ethical limitations. Publicly sharing the transcripts is strictly against general ethical regulations of the Swedish Ethical Review Authority (https://etikprovningsmyndigheten.se/). However, we recognize the importance of data-sharing and to provide insight into the research, and will therefore provide data upon request, evaluating each inquiry individually. For requests concerning data-access registrator@etikprovning.se can be contacted.

**Funding:** This study was funded by Region Stockholm & ALF PPG (grant numbers LS2015-1199, HSNV 11590, and HSN 0904-0396). Author SB has served as an author, consultant, and/or lecturer for Shire/Takeda, Medice, and Roche. He receives royalties for text books and diagnostic tools from Hogrefe, Kohlhammer and UTB. The specific roles of these authors are articulated in the 'author contributions' section. The funders had no role in study design, data collection and analysis, decision to publish, or preparation of the manuscript.

**Competing interests:** The authors have declared that no competing interests exist.

and only participants showing the state of mind profile experienced a decrease of ADHD symptoms accompanied by objectively measured improvements in self-regulation. In addition, compliance affected both how and what strategies were used.

## Conclusion/discussion

A heterogeneous array of cognitive and other strategies is used at varying levels of training compliance by participants with ADHD during SCP that could be condensed to three proto-typical profiles. Future research should take compliance and strategy/training profiles into account when evaluating NF. The latter may help to clarify which and how brain activity regulating mechanisms drive training, individual response to NF, and how they are influenced by motivational factors. Our findings might also help to facilitate more effective instructions in how to approach SCP in clinical practice.

## Introduction

Attention deficit hyperactivity disorder (ADHD) is a disabling and common heritable neuro-developmental condition [1] with a reported prevalence of 5.3% in childhood [2], and around 4% in adulthood [3,4]. ADHD is defined by age inappropriate patterns of inattention, hyperactivity and impulsivity and characterized by executive malfunction, low emotional self-control, and motivational challenges [5]. ADHD increases the risk of diverse impairing outcomes [6], such as failure in academic and occupational careers, social/peer functioning, family conflicts [7–10], criminality, substance use disorder and traffic-related accidents and injuries [11,12]. Furthermore, comorbidity with other disorders is high. Common co-occurring conditions are oppositional defiant disorder, conduct disorder, tics and autism spectrum disorder [13]. Comorbid learning difficulties affect 25% to 40% of children with ADHD [14], and sleeping problems may appear in up to 70% of cases [15].

The most commonly used intervention for ADHD is pharmacological, predominantly using central stimulants (methylphenidate, amphetamines) and noradrenaline reuptake inhibitors (atomoxetine). A recent meta-analysis indicate methylphenidate in children and adolescents, and amphetamines in adults, as preferred first-choice medications for the short-term treatment of ADHD [16]. Still, side effects such as anxiety, emotional lability, headaches, nausea and insomnia are common and occur in between 20% to 50% of medicated children [17–20]. In addition, an equally large percentage of individuals with ADHD show little to no response to pharmacological treatment, or are forced to terminate treatment due to mood instability or hypomania [18,21]. Finally, long-term effects of pharmacological treatment are not well investigated, and concerns have been raised regarding the effectiveness of long-term symptom suppression and long-term side effects, in particular height suppression [22], lower body-mass-index [23] and cardiovascular functioning [24]. Such limitations and concerns provide a rationale for the development and evaluation of non-pharmacological interventions in ADHD. Dietary treatments, such as artificial food exclusion and free fatty acid supplementation, have shown to result in some improvement, similar to cognitive training and neurofeedback (NF) [25].

NF is a form of biofeedback, delivering real-time visualized feedback on a subject's brain activity, most commonly using Electroencephalography (EEG). By applying learning principles such as operant conditioning, NF enables the subject to better regulate brain activity

towards more desired states. There are various forms of NF applications, dependent on which brain activity is targeted. Of the various NF training protocols used in the treatment of ADHD, Slow Cortical Potential NF (SCP-NF) has been considered a preferred standard protocol [26], with some of the strongest evidence for positive effects on ADHD symptoms as well as for medical conditions, such as migraine [27] and epilepsy [28], although other protocols (e.g. sensorimotor rhythm training) have yielded good effects on other conditions as well, such as seizure reduction [29–31]. SCP are a type of event related potentials, measured as slow shifts in the in the bioelectrical activity of the brain. They are characterized by negative or positive shifts lasting from 300 msec. to several seconds [32]. These shifts are believed to reflect states of either increased cortical excitability (negative shifts) or reduced excitability/inhibition (positive shifts) and there are indications that the regulation of SCPs is altered in children with ADHD [33]. SCP-NF aims to increase control over these shifts, ostensibly improving self-regulation and reducing ADHD symptoms.

Although there is a growing body of literature indicating that several NF protocols have beneficial effects in terms of ADHD symptom reduction and enhanced self-regulation of brain activity [34–42], questions remain concerning the efficacy of NF. A systematic review and meta-analysis on non-pharmacological treatment in ADHD by Sonuga-Barke et al. [25] found medium size effects for the efficacy of NF on ADHD symptoms for most-proximal (i.e. parent) measures (Standard Mean Difference (SMD) = 0.59, p < .0001), but these were not endorsed for probably blinded (i.e. teacher) measures (SMD = 0.29, p < .53). A more recent meta-analysis [43] found significant though small size effect improvements from both most-proximal raters (SMD = 0.33, p < .005) and for probably blinded raters (SMD = 0.25, p < .02), when considering inattention symptoms. In addition, there remain issues around NF training requiring clarification, including the reliability of applied outcome measures [41,44,45], the validity of the relationship between NF training and improved symptomatology [46] and the extent to which learning occurs automatically and/or necessitates conscious effort [47]. Furthermore, many subjects are not able to modify their brain activity during NF [48]. In the case of SCP, this subgroup of individuals can be as high as 50% [36,49]. It has been shown though that neurotypically developing children can learn to regulate their SCP within a few sessions [50]. Therefore, research on NF training in ADHD currently desires improved understanding of its mechanisms, mediators and moderators.

In this regard, our current understanding of experiencing and performing NF training is compromised among other things by a scarcity of qualitative research in the field. The few available qualitative studies of lived NF experience have mainly focused on non-clinical adult populations [51,52] and have been limited to descriptions of outcome-related experiences [53], respectively. A theoretical contribution by Gevensleben et al. [47] discussed the degree to which NF challenges the participants' efforts to be efficient vs. acts by automatic learning processes. Depending on which of these actually are dominant or active in NF, different cognitive strategies may have the potential to either avail or hinder treatment effects. A study by Kober and colleagues [54] found that participants using strategies, such as relaxation, concentration and breathing techniques, were outperformed by those not using strategies. In another study, in which NF was used to achieve communication with a locked-in syndrome patient, the authors discussed the possibility that automatic SCP regulation might be hindered by the use of cognitive strategies [55]. While it may be the case that avoiding the use of strategies might lead to better treatment effects, if strategies are applied [56], some might still be more effective or less disturbing than others [50]. Therefore, investigating which types of strategies are used during NF and how these are employed is crucial for the understanding of NF efficacy.

It is debated whether observed NF treatment outcomes can be attributed to specific self-regulatory effects of brain states, e.g. changes in the specific neurophysiological parameters that

are targeted by the NF-protocol in question, or to non-specific NF-related factors, such as positive feedback and interaction with a clinician, or practice effects of sitting still for extended periods of time [41]. General effects, such as those attributable to the client-practitioner relationship, have the potential to influence outcomes directly or indirectly through mediation. An example of mediation would be positive reinforcement from the practitioner that might generate motivation, which in turn may be a prerequisite for learning self-regulation of brain activity. In addition to the aforementioned factors, repetition related and natural factors (e.g. maturation, spontaneous remission), have been discussed in a recent consensus report on NF studies [57].

Placebo effects may also occur in NF, and as Bussalb et al. [58] point out, using probably blinded raters as an estimate for correcting the placebo effect does not appear an appropriate choice, as there is more variability in teacher than in parent ratings. Thibault and Raz [59] refer to a number of studies in which veritable NF effects did not exceed those of sham conditions. Still, a lack of difference in efficacy between sham and "true" NF could in part be due to how sham conditions are designed. A recent study on adults with ADHD that showed no significant differences between NF and sham conditions [60] was critiqued for using automatically adjusting thresholds for reinforcement every 15 sec. at an 80% reward rate. This meant in practice that the mechanisms of operant conditioning were violated, as successful self-regulation lead to increased thresholds, while failure to self-regulate lead to a decreased threshold. Thus, paradoxically, success was punished by increased difficulty, while failure was rewarded by lowering the difficulty level [61]. In this context, it is important to note that both sham and NF actually do show positive treatment effects in ADHD, but the underlying mechanisms for each remain uncertain [59]. Regarding the latter, with a better understanding of participants' experiences, perspectives and strategies of NF training, we could generate novel and testable hypotheses of moderating and mediating variables of treatments effects and mechanisms.

Treatment motivation and compliance with NF-protocols are additional areas that have not been taken into account sufficiently within research on cognitive interventions of ADHD. Still, motivational issues are likely to play a significant role in cognitive training as intrinsic motivation is known to be a reliable predictor of treatment adherence [62]. Various cognitive training procedures have reported positive effects of including game-elements in order to increase motivation within cognitive treatment approaches [63–65]. Evidence suggests that young individuals with ADHD have lower levels of motivation, as indicated by alterations of cognitive-attentional and motivational neural networks [66], reduced self-regulation of motivation [67], as well as low task engagement and performance in the school context [68]. These motivational alterations of individuals with ADHD are likely to be associated with dysregulation of the dopamine pathway [69–73].

In the limited research that is available, the influence of compliance and motivation in cognitive training studies of ADHD is often described in terms of "general participation", coded dichotomously, and not always included in the final data analysis [74–76]. The most frequently used operationalization of motivation and compliance in cognitive training studies of ADHD is a predefined cut-off level of adherence to the treatment protocol, in terms of physically attending the training sessions, and completion of tasks within the sessions. The cut-off point for assuming motivation or compliance is usually set at 75% to 80% of session attendance or task completion [77–80]. However, completion or attendance alone does neither necessarily reflect the motivation, nor the engagement towards the task. Considering the motivational fluctuations in ADHD, due to reduced self-regulation, self-reported motivation may not reflect the actual effort during a session. An expert rating based on a broader definition of compliance, including elements of motivation, could give further insight into how these factors influence the NF procedure.

In conclusion, while NF is gaining increasing attention as a treatment for ADHD, little is known about the participant's subjective experiences during the NF-task. No study has examined NF training strategies, their use and frequency in a clinical population or addressed compliance, including motivational factors and task completion. This is surprising, as issues with self-regulation and motivation are common in ADHD, and may therefore influence the process and effects of NF treatments. The purpose of this study was to describe the use of strategies used by children and adolescents with ADHD during SCP-NF training, while taking the influence of their training compliance into account. This entailed: (a) the mapping of cognitive and other strategies used during SCP-NF training, (b) examining changes over time in these strategies and their relation to training compliance, and (c) investigating strategy profiles in order to gain a more detailed picture of how strategies and compliance are linked and change over time in ADHD

## Method

### Participants

This is a qualitative add-on study to a larger randomized controlled trial of neurocognitive training interventions in ADHD (KITE study; NCT01841151) [81]. The KITE study is a single site randomized controlled comparative trial of neurocognitive training interventions in child and adolescent ADHD, conducted at an outpatient clinical research unit in Stockholm, Sweden. Herein, the efficacy of two types of NF training, slow cortical potential (SCP) and live z-score training (LZS), are contrasted with working memory training (WMT) and a standard care/waiting list control group. Briefly, in the active conditions participants undergo daily training sessions over five consecutive weeks (five sessions per week, 25 sessions total). Each SCP session consisted of 144 trials, split into four blocks that lasted for 10 sec. (2 sec. baseline, 8 sec. feedback). Two booster sessions before a six-month follow-up assessment were also included (see Hasslinger et al., 2016, for a complete study description). All procedures were approved by the Ethical Review Board in Stockholm. Two-hundred children and adolescents aged 9 to 17 years are included in the KITE study. All had received a primary diagnosis of ADHD (ICD-10: F90.0; DSM- IV-TR: 314.00, 314.01). They were recruited either via self-referral or as clinical referrals via compulsory mental health care providers for youth in Stockholm County (child and adolescent psychiatry, pediatrics). Common neurodevelopmental comorbidities such as autism spectrum disorder, learning disabilities and language impairments were no exclusion criteria, whereas the presence of unstable disorders that could shift during the study (e.g. eating disorders, bipolar disorders) lead to exclusion, as did an IQ < 80 and limited understanding of basic Swedish language. Ongoing pharmacological treatment of ADHD was allowed in the KITE study, the dosage had to remain stable though during the study. Participants received a gift certificate of SEK 200 (USD ~22) after training completion and an additional certificate worth SEK 500 (USD ~55) after completing follow-up assessments.

A subsample of 30 participants (mean age M = 12.41, SD = 2.73), 9 girls and 21 boys from the KITE study, participated in the present qualitative study. The sample consisted primarily of participants who had been randomized into the SCP group (n = 21). However, we also included participants that originally were included in the WMT or waiting list group, and who after their completion in the study were offered 25 sessions of SCP (n = 9) for ethical reasons. Sample characteristics are summarized in Table 1.

### SCP task

For the SCP the THERA PRAX-qEEG amplifier (neuroConn GmbH, Ilmenau, Germany) was used. Central zero (Cz) was used for the EEG electrode, with the mastoids serving as ground

**Table 1. Sample characteristics for the total sample and compliance subgroups.**

| | | Compliance | | |
|---|---|---|---|---|
| | Total sample | Low compliance | Medium compliance | High compliance |
| N | 30 | 2 | 16 | 12 |
| Age years (M, SD) | 12.41 (2.73) | 11.28 (1.01) | 12.20 (2.91) | 12.89 (2.72) |
| Sex (male:female) | 22:9 | 1:1 | 13:3 | 8:4 |
| Completed qualitative interviews | 130 | 8 | 60 | 62 |
| IQ (M, SD) | 102.7 (16.7) | 102.5 (17.5) | 102.2 (18.7) | 103.5 (13.3) |
| ADHD—severity (M, SD) | 18.68 (6.23) | 20 (4.24) | 19.24 (7.24) | 17.67 (5.09) |
| Comorbidity ASD/Other | 5/11 | 0/1[1] | 2/5[2] | 3/5[3] |
| ADHD type (ADHD/ADD)[4] | 23/7 | 2/0 | 12/4 | 9/3 |
| Medication | 14/30 | 0 | 7/16 | 7/12 |

ADHD-index Conners 3 parent report version (max. 30);

[1]Depressive episode, speech disorder;

[2]Generalized anxiety disorder (GAD), motor developmental disorder, sleep disorder, speech disorder, reading disabilities, ODD;

[3]Separation anxiety, Specific phobia, GAD, Obsessive-Compulsive disorder, Oppositional defiant disorder, reading disability;

[4]ADHD = combined type, ADD = ADHD predominantly inattentive sub-type.

(left) and reference (right). In addition, four electrodes were placed around the eyes to measure the vertical and horizontal electrooculogram (EOG). Ag/AgCl electrodes were used at all sites, and impedance was kept under 5k Ohm. Prior each session the subjects eye movements were calibrated for an online correction, eliminating or suppressing signals from the eyes. Signal changes that exceed 200μV, where automatically rejected and retaken, as these indicated artifacts.

During the task, the subject was presented with a triangle on the screen pointing either up or down and was asked to intentionally try to steer a second object appearing on the screen in the same direction as the arrow [82]. This was done by attempting to control his or her cortical activity by voluntarily creating negative or positive slow cortical potentials. In addition to the visual prompting (triangle pointing up/down) there was also an acoustic prompting in the form of two different sounds. Each trial lasted 8sec, and was preceded by a 2sec-baseline calculation, and if successful a visual reward (a sun) was displayed (Fig 1). The reward was only displayed if the participant could steer and remain above a threshold that was set at 0–40 μV.

Every session was split into four blocks, each consisting of 36 trials (18 trials up, 18 trials down). Furthermore, some trials were so-called transfer trials. These did not include any active feedback during the trial, though they did bid the reinforcement when the trial was successful. These transfer trials served to facilitate the self-regulation beyond the need for real-time

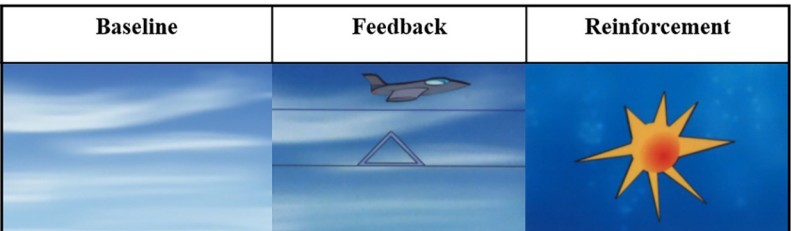

**Fig 1. SCP feedback.** Baseline is calculated for 2s, followed by and 8s feedback. If the object (airplane) is kept in the same direction as is prompted (where the triangle points), a sun is displayed (reinforcement).

feedback. The proportion of transfer trials were 20% during the first week, 40% during the second week and 50% during the remaining three weeks of SCP training.

The SCP training was framed as a treatment for ADHD, by improving one's attention regulation. However, due to the participant's varying age and level of awareness regarding their ADHD diagnosis, framing information was adapted to the individuals' prerequisites. Participants were instructed to sit still, try to be physically calm and to avoid unnecessary movements. During the sessions, participants were instructed to try to steer the object in the correct direction, and get as many rewards (suns) as possible. Instructions and "coaching" during the sessions were tailored to individual requirements. For example, in some cases, the focus was on prompting the participant to avoid physiological artefacts, while in others continuous input on the number of remaining trials was given, while other demanded quietness.

## Qualitative interviews

Parallel to the NF training, we conducted a series of repeated short semi-structured interviews, in order to investigate the use of cognitive and other strategies in SCP. The interviews were guided by several open-ended questions that requested the participants to describe in their own words individual experiences of SCP, and particularly their use of strategies for successful NF task completion. By repeating the interviews throughout the NF training, we encouraged the participants to reflect on and develop their answers. It also gave us the opportunity to track changes over time, and enabled us to examine the use of strategies throughout the NF process. The interviews explored the participants' subjective experiences ("What do you think about the training?" and their use of potential strategies ("How did you do it?"). The section of the interviews that dealt with subjective experience of training were initiated by a general question, about how the participant had perceived the overall training situation, followed a more detailed and response-guided inquiry of the personal experiences of training. Subsequent questions concerned the approach to, and potential use of strategies during training (combined with more specific, clarifying questions regarding those strategies when needed). After the final session, as well as after the booster sessions, participants were also asked to report on experiences of perceived training effects. The length of the interviews varied. The mean interview length was M = 2:52 min. (Mdn: 2:46 min.; Min: 0:21 min.; Max: 8:57 min.). Interviews were audio recorded using a digital audio recorder (H4n, ZOOM Corporation, Japan).

The interviews were administered immediately after the first SCP training session. The interviews were then repeated after each fifth session (i.e. session 5, 10, 15, 20, 25), as well as after the booster session that were scheduled six months after the participants had finalized the SCP training. Members of the research team, who had also conducted the preceding training session, conducted the semi-structured interviews. All interviewers had relevant clinical training, and experience of NF-training within the research project. The team of trainers and subsequent interviewers consisted of one certified clinical psychologist, one research nurse, and four research assistants who were in the final years of training for their clinical psychologist degree at a Swedish University.

## Compliance rating

Participants' compliance was rated on a 0 to 3 scale by the trainer directly after each training session, where "0" represented a complete lack of compliance and "3" high compliance. The compliance rating aimed to capture the participants' adherence to successfully perform the SCP task as intended. Hence, compliance was operationalized in a broader sense to cover both attendance and motivation, but also elements of how well the task was completed. A participant could be highly motivated and still get a low compliance rating, or be lacking motivation

but be rated high on compliance, as the rating was based on an overall assessment of the participant. Moving, talking and other activities that interfered with the task (or the quality of the EEG signal) were elements that lowered the rating, while determination to succeed, and conformity to instructions would increase the rating. Considering the daily session ratings, an overall rating was generated via consensus discussion by the trainers. Hence, the final rating is not an average of the daily session rating, but an overall appraisal of the participants' SCP compliance.

## Data analysis

Audio recordings of the interviews were transcribed verbatim and then coded into consistent emergent themes aided by NVivo 12 (QSR and Ltd, 2012), using thematic analysis, implementing the six-step process as described by Braun and Clarke [83]. After an initial coding of semantic content, i.e. explicit components of the text, the data was reorganized into meaningful groups, based on material concerning the use of strategies, hence filtering out material that was not relevant. The material was then sorted according to task (i.e. for SCP steering the object up/down). At the next stage, all coded data was analyzed, and the coded extracts were sorted into potential themes. Thereafter, the candidate themes were reviewed, named and organized into a hierarchical structure. Finally, the material was merged and the themes were reanalyzed and interpreted to generate a broader contextual understanding, and the material was thematically interpreted, named and structured based on consensus by two to three researchers. All available interviews from all participants were coded, even if they only completed a single interview.

The analysis was focused at a semantic level, however, studying the material as a whole, elements on a latent level emerged as well [83]. The latent themes were reflected in most semantic themes, but not all, and not in every code within the theme. While the semantic themes were mutually exclusive, the latent themes were more similar to *themes* described in Qualitative Content Analysis [84], where *themes* have a quality of linking together the underlying meanings of categories, which in this study would be the semantic themes. For clarity, the semantic themes will be referred to here as themes and sub-themes, while the latent themes will be referred to as latent themes. As the semantic themes solely entail the description of strategies, these terms will be used interchangeably.

For the analysis of change over time on strategy use, we examined the presence of the emerged themes and sub-themes per interview, thereby generating the frequency of each theme per session. Four members of the research team evaluated strategy changes over time based on visual inspection of the themes. After individual inspections, observations were discussed, until consensus was reached. The participants were then separated based on their compliance rating. Since only two participants where rated into the low compliance category, they were merged with the neutral compliance group into a Neutral Compliance (NC) group. In order to facilitate the analysis, we grouped sessions into three periods: early (sessions 1–5), middle (sessions 10–15) and late sessions (sessions 20–25).

Finally, for the in-depth analysis of strategy profiles, participants who had completed at least five of the six interviews (excluding booster session interviews) were selected, which resulted in 14 individuals. The complete interviews from each session were condensed, which served as basis for summaries of every participant. The interviews were reread and compared to the condensed versions. Notes were taken for the complete sets of interviews. Based on the notes, summaries and the previously found categories/themes, the material was analyzed for patterns for emerging profiles, via consensus discussions, and included the use of strategies as well as the participant's motivation and attitude towards the training. Comments concerning

experienced outcome were also considered. Once a clear description for an individual had been established, this was then used as a comparison description for the other interview sets. Hence, we used an inductive-deductive approach in this analysis.

## Results

### Strategies used

Four domains of strategies emerged: **Cognitive Strategies (C)**, **Emotional Regulation Strategies (E)**, **Physiological strategies (P)** and **Unspecified Strategies (U)**. Each domain consisted of a number of ***themes***, and some themes were composed of a number of *sub-themes*. In addition, an overarching latent theme, ***Arousal Regulation (AR)***, emerged. AR is involved in most strategies, and can be described as a general mechanism that the different strategies are aiming to influence. AR consists of two sub-themes, *Increased Arousal* and *Decreased Arousal*, which can be present concurrently in some strategies, especially within the Cognitive and Emotional domain (e.g. *visualizations* and ***Specific Emotions***, which can both be used to increase or decrease the subject's arousal level). However, AR was also present in themes within the Physiological domain, e.g. using ***Decreased Activity*** to decrease one's arousal level. An overview is provided in Fig 2.

 **Cognitive strategies.** This domain emerged as most coded domain and consisted of six themes: ***Focus strategies***, ***Generating Internal Phenomena***, ***Memory Recall***, ***Motivation***, ***Thought Avoidance***, and ***Wakefulness***, with three themes having sub-themes. The themes in this domain describe strategies that all attempt to modulate cognitive processes, such as attention, memory, inhibition of thoughts or strengthening motivation. In the ***Focus*** theme participants attempted to alter their attentiveness by shifting their focus in a particular direction. It embraces three subthemes (*directed focus*, *scattered focus* and *concentration*), reflecting different nuances of focus. The ***Generating Internal Phenomena*** theme included strategies such as the visualization of both mental imagery and the desired outcome (*visualization*), the generation of internal sounds and commands (*auditory imagery*) and direction-oriented thinking (*thinking in a direction*) to steer the object on screen. Certain participants attempted to control their performance by bringing to mind autobiographical events or semantic information (***Memory Recall***). Strategies in the ***Motivation*** subtheme aspired to drive the participant to work towards the desired goal and included uttering words of encouragement to oneself as well as persisting despite previous unsuccessful attempts to control the stimulus on screen. ***Thought Avoidance*** strategies entailed active attempts to suppress thoughts or abstain from thinking as a whole. ***Wakefulness*** strategies attempted to regulate one's level of wakefulness to achieve the desired outcome. They could fall into either the subtheme "*sluggishness*", where the participants tried to deaccelerate a thought processing to a slow cognitive tempo as a way to control the stimulus on screen, or "*alertness*", where they would attempt to control the stimulus by enhancing their thought processing and mental load. Quotes defining each theme and sub-theme are provided in Table 2.

 **Emotional regulation strategies.** This domain was divided into three themes: ***Specific Emotions***, ***Mindfulness*** and ***Emotionally Engaging Thoughts***. The first theme pertained to strategies involving the use of specific emotions, such as happiness, sadness and anger, to elicit control over training outcome. The ***Mindfulness*** theme concerned strategies whereby the participants attempted to reach a meditative state of serenity and harmony by thinking calming thoughts and focusing on one's breathing. Lastly, strategies in the ***Emotionally Engaging Thoughts*** included thinking about things that elicited strong emotions, such as "dangerous" things, one's favorite game or moral and political questions. Table 3 provides quotes defining the three themes.

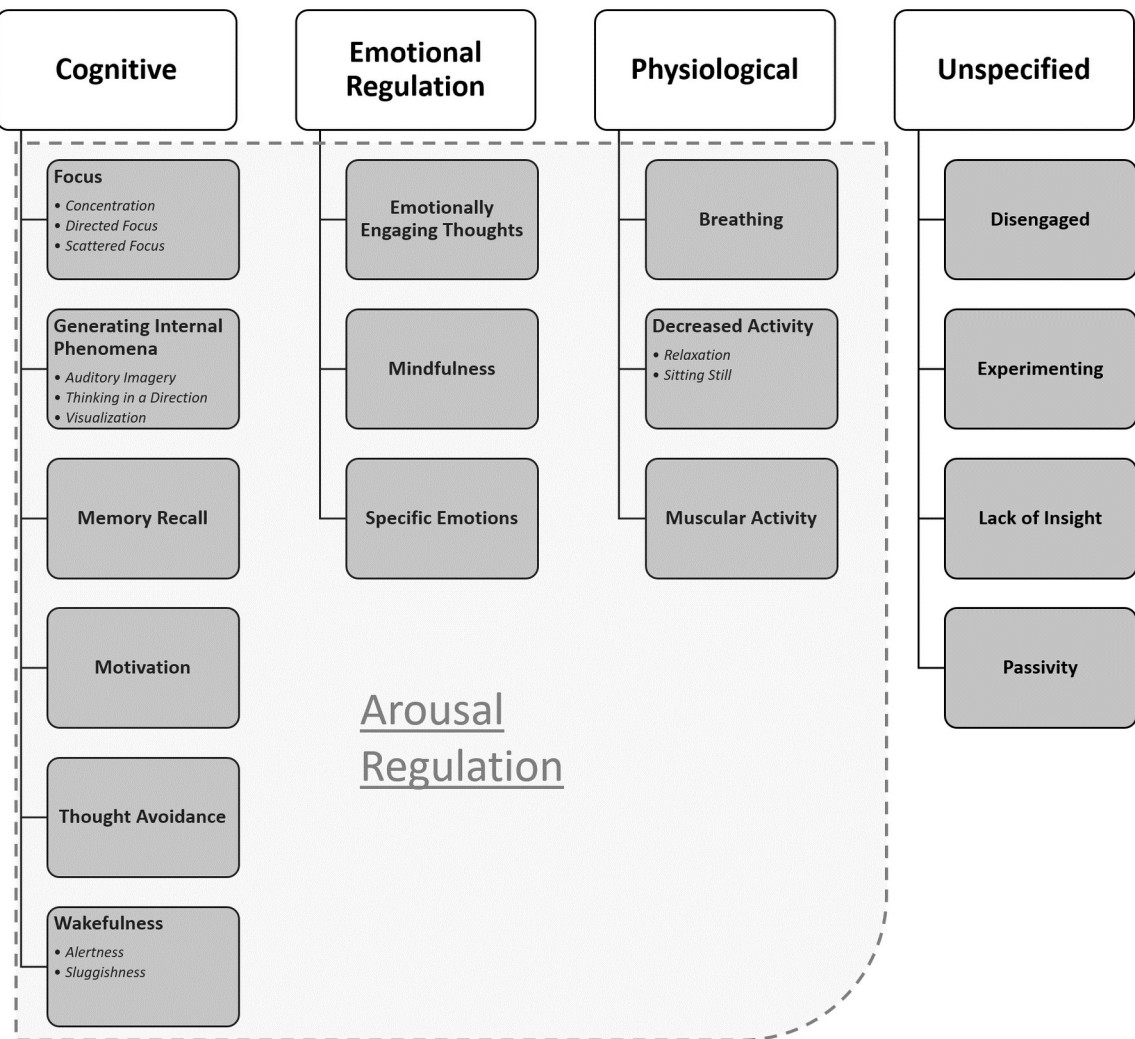

**Fig 2. Hierarchical overview of themes and sub-themes.** The above figure shows the four identified domains: **Cognitive, Emotional Regulation, Physiological** & **Unspecified**. Below each domain themes are illustrated as boxes, with sub-themes inside the box. The dotted line illustrates the latent themes (Arousal Regulation) and where it overlaps with the other themes.

**Physiological strategies.** This domain featured either a decrease in physical activity or an increase in muscular activity, and consisted of three themes: *Breathing, Decreased Activity* and *Muscular Activity*. *Breathing* describes activities that used breathing to control the shifts, i.e. breathing heavily or holding ones breathe. *Decreased Activity* was divided into two sub-themes: *sitting still* and *relaxation*. In order to differentiate from the emotionally driven relaxation in *Mindfulness* and the hypo-activity of *sluggishness*, the physiological subtheme *relaxation* referred exclusively to strategies in which participants attempted to induce muscular relaxation without necessarily striving towards a mental relaxation. The *Muscular Activity* subtheme encompassed strategies where participants attempted to move or tense body parts. See Table 4 for quotes indicating the themes.

**Unspecified strategies.** This domain contained strategies that did not fall within any of the aforementioned categories. Strategies in this domain were divided into the following themes: *Passivity, Disengagement, Experimenting* and *Lack of insight*. The theme *Passivity* was defined by abstaining from active behaviors as a mean of achieving the goal, it includes

**Table 2. Quotes on cognitive strategies during slow cortical potential training.**

| Theme/Sub-theme | | Quote |
|---|---|---|
| **Focus**[1] | C1 | |
| *Concentration*[2] | C1.1 | **Session 10 (boy, 13 years)** |
| | | "So you should be very focused when it should go up. And like only think about the stork or whatever you have." |
| *Directed focus*[2] | C1.2 | **Session 1 (girl, 10 years)** |
| | | "I focused on the star." |
| *Scattered focus*[2] | C1.3 | **Session 10 (boy, 13 years)** |
| | | "Yeah, hard to explain but I can think like, for example maybe about two things at the same time (. . .) you should get as many things in the head as possible and then. . . it goes down." |
| **Generating Internal Phenomena**[1] | C2 | |
| *Auditory Imagery*[2] | C2.1 | **Session 1 (boy, 17 years)** |
| | | "Yes, but not so much up. I can't think of anything like that, but down, it kind of worked to think like this "ha ha ha ha" kind of, like in dark tones that sound pretty much in minor" |
| *Thinking in a direction*[2] | C2.2 | **Session 5 (boy, 9 years)** |
| | | "I thought that when it should go down, I thought in my head down, and when up, I thought up." |
| *Visualization*[2] | C2.3 | **Session 20 (boy, 17 years)** |
| | | "I was creating landscapes, and then I thought I would make two pieces that are very opposite to each other to try to have any effect of one going down and the other going up. Ehm, where one was like this .. some black island with lots of lava all around and everything is dead and misery and stuff .. and the other is some kind of rain forest and .. everything is green and it wedges past any rat or anything .. like life like this .. and it works pretty well." |
| **Motivation**[1] | C3 | **Session 1 (girl, 16 years)** |
| | | "I think that when it's supposed to go up then you should just like, then I think things like "come on now, you can do this, it will go up", like you focus more." |
| **Thought Avoidance**[1] | C4 | **Session 25 (girl, 16 years)** |
| | | "And when it's supposed to go down then I try and think of like nothing. Really (think) that I'm in a room and it's dark and there is nothing, like. It normally works." |
| **Memory Recall**[1] | C5 | **Session 5 (boy, 17 years)** |
| | | ". . .I tried to . . .it worked to think about old, old memories, it worked quite well." |
| **Wakefulness**[1] | C6 | |
| *Alertness*[2] | C6.1 | **Session 15 (girl, 16 years)** |
| | | "And then when it shall up, then I think that I have to be sharp there, like you know, like being ready for anything." |
| *Sluggishness*[2] | C6.2 | **Session 5 (boy, 9 years)** |
| | | "And when I should go down, I should be a bit sleepy." |

[1]Themes;

[2]Sub-themes

activities such as unspecified thought and mind wandering. Strategies in which the participant abstained from action due to a lack of interest in the activity or its results fell under the ***Disengagement*** theme. When participants were unaware or uncertain about the strategy they used during training, their answers were coded under the ***Lack of insight*** theme. Lastly, the ***Experimenting*** theme included strategies that followed the principle of trial and error and were employed foremost by participants who lacked a regular strategy and instead tested

**Table 3. Quotes on emotional regulation strategies during slow cortical potential training.**

| Theme | | Quote |
|---|---|---|
| Emotionally Engaging Thoughts | E1 | **Session 11 (boy, 13 years)** |
| | | "It helps if one thinks . . . like, about something that you like doing. But, and really focus on that." |
| Mindfulness | E2 | **Session 5 (boy, 17 years)** |
| | | "I let a thought. . . if a thought flows in, I just let it keep on flowing away again." |
| Specific Emotions | E3 | **Session 10 (girl, 16 years)** |
| | | "Things that make me, like, happy . . . something that makes me really hyped." |

different ones during the course of training. Quotes indicative of each theme and sub-theme are exemplified in Table 5.

## Strategy changes over time

**Changes over time—General.** Descriptively, we found some mentionable changes over time during training in the Cognitive domain. *Focus* increased after the first week (after session 5) of training, in particular regarding its sub-theme *directed focus*. In addition, *Wakefulness* showed an increase during the first three weeks, before staggering towards the end again. The other themes remained widely stable over time. Within the domain of Emotional Regulation, the frequency of strategies did not change over time, though *Emotionally Engaging Thoughts* was more common during the booster sessions compared to the other sessions. In the domain of Physical strategies, we found that *Breathing* was only prevalent during the first week, and then not mentioned any more, while *Decrease Activity* was reported more towards the end of the training. *Experimenting* was reported most frequently during the first week, similar to *Lack of Insight*, which lasted for two weeks before decreasing. Apart from the above-mentioned, clear patterns or trends were scarce, when looking at the material as a whole. The frequencies of how often a theme was mentioned is presented in the supporting information S1 Table, and separated for every session.

**Strategy changes over time in relation to compliance.** When comparing High Compliance (HC) with the Neutral Compliance (NC) group, some obvious descriptive differences and changes over time emerged. *Focus* and *Wakefulness* were more common overall and earlier in the HC-group, than in the NC-group, though no difference for *Wakefulness* at booster.

**Table 4. Quotes on physiological strategies during slow cortical potential training.**

| Theme/Sub-theme | | Quote |
|---|---|---|
| Breathing[1] | P1 | **Session 5 (boy, 17 years)** |
| | | ". . . just like take deep, calm breaths." |
| Decreased Activity[1] | P2 | |
| Relaxation[2] | P2.1 | **Session 25 (girl, 14 years)** |
| | | "I try to like relax my body a little" |
| Sitting still[2] | P2.2 | **Session 5 (boy, 12 years)** |
| | | "It's just to sit. . . calm and still" |
| Muscular activity[1] | P3 | **Session 5, (girl, 11 years)** |
| | | "I move the tongue. . . sideways." |

[1] Themes;

[2] Sub-themes

**Table 5. Quotes of unspecified strategies during slow cortical potential training.**

| *Theme* | | Quote |
|---------|---|-------|
| *Disengagement* | U1 | **Session 20 (boy, 11 years)** |
| | | *"I don't even try. . ."* |
| | | **Session 25 (boy, 10 years)** |
| | | *"Sat here and didn't do shit!"* |
| *Experimenting* | U2 | **Session 5 (girl, 16 years)** |
| | | *". . .it's different, I'm like trying out."* |
| *Lack of Insight* | U3 | **Session 10 (boy, 10 years)** |
| | | Participant: *"It feels like I'm getting more suns (i.e. successful trials)"* |
| | | Trainer: *"Ok. . . and how are you doing that?"* |
| | | Participant: *"I don't knoooow! I guess I think. . .. In some way. . . I'm not sure!"* |
| *Passivity* | U4 | **Session 10 (boy, 9 years)** |
| | | *". . . I just let it move around. Then I don't concentrate very much."* |

**Generating Internal Phenomena** was slightly more common in the NC-group, though no difference were reported at booster. **Emotionally Engaging Thoughts** were almost exclusively reported in the HC-group (except for the first week). **Mindfulness** was reported earlier in the HC-group, though **Specific Emotions** were more common in the NC-group.

Overall, at booster, half of the HC-group reported using or having used strategies within the domain of Emotional Regulation, compared to 13% in the NC-group. There was a higher increase in using **Decreased Activity** as a strategy in the HC-group compared to the NC-group, while **Muscular Activity** was used increasingly in the NC-group. This was especially clear at booster (0% vs. 63%). **Passivity** more common for the NC-group in the later training sessions, and especially at booster (13% vs. 50%). An overview over the differences in strategy use between HC and NC is presented in Table 6. Sessions are presented in pairs, representing the early, middle and late phase of training.

## Strategy profiles

During the in-depth analysis of the complete interview-sets, we found three profile types. These profiles consisted of patterns of identified strategies, as well as motivation and perception of the training. Six subjects predominantly described strategies that manipulated their *"State of Mind" (SM)*, four subject confronted the SCP task in a *"Manifest and Concrete" (MC)* fashion, while the remaining four subjects gave an overall impression of being somewhat *"Unaware" (Un)* of what they were doing.

Beyond the differences in strategic approach, subjects in the three strategy-profiles also differed on age (mean age 13.67y (SM), 11.2y (MC) and 10.45y (Un)), symptom severity based on Conners 3 parent-report questionnaire (mean score 19.83 (SM), 22.75 (MC) and 21.25 (Un)) and the compliance rating (2.67/3 (SM), 2/3 (MC) and 2.25/3 (Un)). An overview with summaries for all subject's interviews is available in supporting information S2–S4 Tables.

Although this is a qualitative study, in order to complement the participants' subjective experiences with an objective measure, we also sought to link strategy profiles to actual performance during SCP. For this, we calculated the average μV value of the last 3 sec. for each direction (i.e. activation/deactivation) and conditions (i.e. real-time feedback/transfer), which were calculated in the TheraPrax™ for each block. The values were then plotted for each of the three strategy profiles, and are depicted in Fig 3. The graphs in the first two columns show the performance during activation (-) and deactivation (+) for real-time feedback (FB) and transfer trials (TR). The column on the right illustrates the difference between—and + for the FB and

**Table 6. Changes over time in strategies in high vs. neutral compliance—Frequency of strategies present in interviews per total interviews.**

| | High compliance | | | | | Neutral compliance | | | |
|---|---|---|---|---|---|---|---|---|---|
| Early | Middle | Late | Booster | | | Early | Middle | Late | Booster |
| **61%** | **69%** | **76%** | **50%** | **Cognitive** | | **56%** | **67%** | **72%** | **63%** |
| 22% | 50% | 41% | 50% | Focus | | 7% | 17% | 39% | 13% |
| 28% | 25% | 35% | 25% | GIP | | 37% | 28% | 50% | 25% |
| 0% | 13% | 6% | 0% | Memory Recall | | 4% | 0% | 6% | 0% |
| 6% | 6% | 6% | 13% | Motivation | | 0% | 6% | 6% | 13% |
| 6% | 13% | 18% | 13% | Thought Avoidance | | 7% | 17% | 0% | 25% |
| 28% | 50% | 24% | 13% | Wakefulness | | 7% | 11% | 6% | 13% |
| **28%** | **50%** | **29%** | **50%** | **Emotional Regulation** | | **26%** | **22%** | **33%** | **13%** |
| 0% | 19% | 6% | 25% | EET | | 7% | 0% | 0% | 0% |
| 28% | 38% | 12% | 25% | Mindfulness | | 7% | 0% | 22% | 13% |
| 0% | 6% | 12% | 25% | Specific Emotions | | 15% | 22% | 17% | 0% |
| **33%** | **19%** | **35%** | **38%** | **Physiological** | | **22%** | **28%** | **39%** | **75%** |
| 17% | 0% | 0% | 0% | Breathing | | 4% | 0% | 0% | 0% |
| 6% | 6% | 35% | 38% | Decreased Activity | | 7% | 0% | 22% | 25% |
| 17% | 13% | 6% | 0% | Muscular activity | | 19% | 28% | 17% | 63% |
| **50%** | **44%** | **29%** | **25%** | **Unspecified** | | **41%** | **61%** | **50%** | **63%** |
| 6% | 0% | 18% | 0% | Disengaged | | 4% | 17% | 17% | 13% |
| 22% | 6% | 6% | 13% | Experimenting | | 15% | 6% | 6% | 13% |
| 11% | 19% | 6% | 25% | Lack of Insight | | 11% | 11% | 11% | 25% |
| 22% | 25% | 6% | 13% | Passivity | | 19% | 33% | 33% | 50% |

Early = sessions 1 and 5, Middle = sessions 10 and 15, Late = sessions 20 and 25, GIP = Generating Internal Phenomena, EET = Emotionally Engaging Thoughts

TR trials (i.e. difference between FB+/TR+ and FB-/TR-). Row (a) shows the results for the "State of Mind" profile, indicating differentiation during FB, and to some lesser extent for TR. Regarding the differences for +/-, a trend for an improvement revealed for FB, and to a lesser degree for TR trials. For the "Manifest and Concrete" strategy profile (row b), a clear differentiation both for FB and TR emerged. However, improvements over time were small in "State of Mind" strategy profile for FB and even decreased for TR. The "Unaware" strategy profile showed no consistent pattern, with no indication of improvements in differentiation of FB or TR over time. Raw data is provided in S5 Table.

**State of mind.** This profile is characterized by task solving behaviors such as manipulating one's mental state, most commonly via shifts in focus and/or wakefulness. Almost all strategies were within the Cognitive Domain, especially containing focus and manipulation of "State of Mind". Though different strategies were used over time, these seemed to be mostly progressions of earlier described strategies rather than novel ones. Overall, the pattern of strategies was stable overtime. The most frequently reported strategies were *Focus* (51%), *Wakefulness* (49%), *Mindfulness* (34%) and *Generating Internal Phenomena* (34%). In this profile, the motivation seemed intrinsic, and even if the subjects did not enjoy the task, they still knew that there was a benefit for them and they did their best, and the improvements after training were reported as self-perceived. Most participants in this profile were astute in their strategy description, and are able to elaborate their statements, as is exemplified by the following case descriptions.

ADHD combined, 13y, boy: This boy applied the same strategies Focus and Wakefulness, more or less throughout all of the 25 sessions. He described the main strategy as thinking

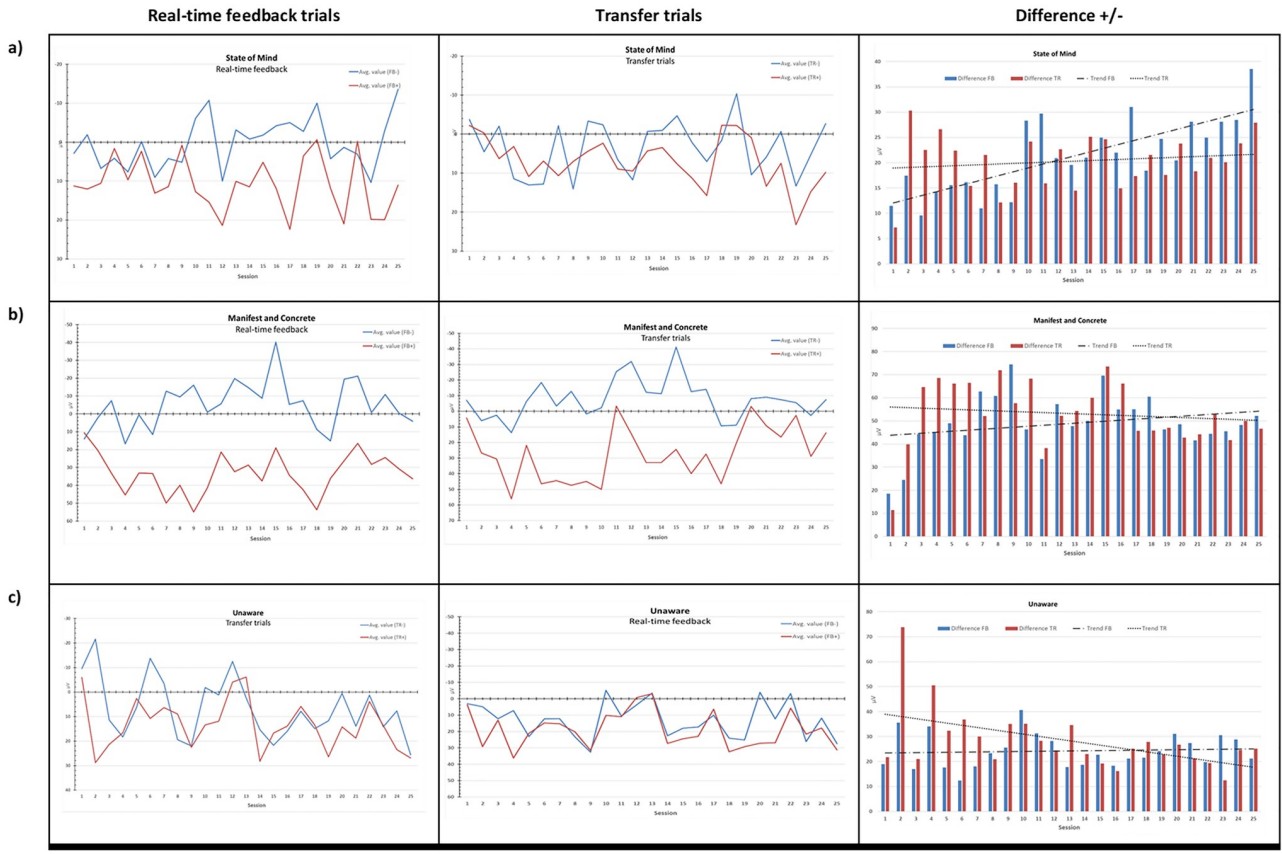

**Fig 3. SCP regulation profiles.** Graphs illustrating the performance during real-time feedback (FB)[left] and transfer (TR)[middle] trials. FB- and TR-indicates the performance during activations trials (i.e. increased negativity) and FB+ and TR+ indicated performance during deactivations trials (i.e. increased positivity). Difference +/- (right) shows the difference between activation trials (FB- & TR-) and deactivation trials (FB+ & TR+). Trend lines for FB and for TR are also shown. The average performance and average differences per session for participants in the "State of Mind" profile are shown in a), the "Manifest and Concrete" profile is shown in b), and the "Unaware" profile is shown in c).

"slow and steady" when attempting to get the object to go upwards, and "fast and steady, such as thinking about different things at the same time", when attempting to get the object in the SCP task to move downwards. Thinking "slow and steady" can comprise visualization of peaceful scenes and thinking "fast and steady" may consist in visualizing lively and active imagery. The participant reported progressively higher levels of control over training results as the weeks went on as well as a perceived improvement of symptoms in everyday life, especially in regards to concentration. Attitude towards training was positive throughout.

> *I have the tactic that you should . . . . when you want (to steer the object) upwards you should be calm and think slowly and be very concentrated. . . be concentrated on one thing. While when going down, you should be focused on several different things and think about it quickly. . . for example, . . . some leafs that blow in the wind or something calmer, but then when it goes down, I maybe think of like three different cats that maybe run around. . .*

ADHD inattentive subtype plus ASD, 17y, boy: The most frequent strategies used by this participant were ***Focus***, ***Emotionally Engaging Thought*** and ***Memory Recall***. This boy was highly motivated and compliant. He reported a variety of strategies within the cognitive domain. He reported difficulties to steer the object up or down in the SCP task, but had

identified the different states required to succeed in the task, and can describe them somewhat. He reported improvement concerning his concentration and mental stamina, especially considering academic activities:

> *I'm sure I've gotten better concentration because I've been able to sit and work and, or study, (. . .) and devote myself to something that I'm not so keen to do, like typing a 600 word essay (. . .) I can sit for a long time at a time and be focused on it without leaving and doing other things and stuff.*

> *Yes, I feel that I can listen to (. . .) that I can listen to something completely focused much longer than I was able to do before without, like, start thinking about other things.*

**Manifest and concrete.**    Here, participants approach the task in a manifest or concrete way. The strategies are often from the Physiological domain, while the Cognitive strategies that are mentioned are described are less specific such as "*thinking in a direction*". Motivation and compliance for the task can be high, however, the motivation seems extrinsic, stemming from external rewards. Most improvements that were reported in this profile, originated from teachers or parents, while self-perceived improvements were scarcely mentioned. The most frequently reported strategies were **Passivity** (48%), **Muscular Activity** (43%) and **Generating Internal Phenomena** (35%). The concrete approach within this profile is illustrated by the cases below.

ADHD combined, 10y, boy: This participant had a more concrete approach towards the training, and was using **Muscular Activity** throughout the training. His use of **Generating Internal Phenomena** involved predominantly the subtheme *thinking in a direction*, which is less specific compared to the other subthemes (*visualization* and *auditory imagery)*, and hence leaves uncertainty concerning the actual activity. No major shifts in the use of strategy were observed over the SCP training period. His sense of control over the task varied substantially, from total to no experienced control.

> Interviewer: *"Okay, (. . .) can you feel that you have trained/improved something?"*

> Participant: *"The brain.. but. . . I don't know if one can notice it?"*

> Interviewer: *"No, exactly, (. . .) how do you notice it?"*

> Participant: *"Erh, I don't know how I notice it."*

ADHD combined, 12y, boy: This participant showed a low level of motivation and compliance. He reported mainly strategies within the Physiological domain and the Unspecified one, regulating the signal via muscular activities and/or via passivity and disengagement. This participant complained about missing other activities due to the daily training sessions. At follow-up, this participant reported improved school grades. These improvements were based on his teachers' opinions, though he himself did not register noticeable changes:

> *No. But I have become more concentrated in school and so (. . .) Ehh. . . my grades. . . have gotten better too (. . .) and then my teachers have said that "You don't give up" and that I have become better at . . . listening and so on.*

**Unaware.**    This profile was found in participants who did not exhibit a particular approach in handling the task. Often many different strategies were tried, not settling for any specific strategy. Common among participants showing this profile was, however, a lack of insight in

what they were supposed to be doing. Hence, strategies from the Unspecific Domain were reported frequently. The most frequently reported strategy was ***Lack of Insight***, all other strategies were reported less frequent, showing that many strategies were tried but never implemented over time. The cases below illustrate this.

ADHD and ASD combined, 10y, boy: This participant made clear that he was solely motivated by the compensation (gift card) that he would be getting at the end of the study. He had difficulties to describe what he was doing, but reported trying different strategies from all domains, however, none was used systematically over time. Furthermore, he did not seem certain of what was expected of him during the task, nor did he notice any effects or improvements from the training. Nonetheless, he did have a positive attitude towards the training.

> Participant: *"At some point I'm sure I'll think it's fun, but not right now when I don't understand shit!"*

> Interviewer: *"Is that how it feels right now? That you don't understand shit?"*

> Participant: *"It's a little fun but it's still . . . I don't understand what I'm supposed to doooo!"*

> Participant: *"Hm . . .how did you do to steer it up or down?"*

> Interviewer: *"I don't know (laughter). It just happens. . ."*

ADHD combined, 10y, boy: This participant started with Physical strategies, that shifted over to Unspecified and unclear strategies that could not be fully described. After two weeks, the participant did not know what he was doing, and described mainly a trial-and-error approach. After three weeks, the participant described ***Emotionally Engaging Thoughts*** as his main strategy, however at the same time he describes it as ineffective and therefore did not use it tenaciously. During the fourth and fifth week, again, the participant could not describe that he applied any particular strategy. His main goal was to sit still to avoid artefacts. Even though this participant did describe several strategies, no strategy was consistently applied. The participant's strategy-profile was dominated by a lack of insight, and a passive frustration. Though he gave the impression of complying well with the task, his motivation dwindled.

> *You get used to it. . . at first you find it very fun, whatever. Then it just gets more and more boring for me. . .*

## Discussion

The current study shows that individuals use a wide array of cognitive and other strategies during SCP-NF training: 16 strategy themes and 11 sub-themes were identified and later sorted into four generic thematic domains: physiological, cognitive, emotional regulation and unspecified strategies. These four domains are similar to the three strategies (*muscle contraction*, *concentration* or *preparation for pushing a button*) that Siniatchkin et al. [50] used in their questionnaire, when asking neurotypical children about their strategy use, and those found in neurotypical adults (Concentration, movement, arousal, focused vision, Emotion and imagery) [56].

When coding and structuring the verbal material to themes, we focused on the semantic meaning of the text, and several elements had to be interpreted and extrapolated, as clearer descriptions by the participants were scarce. This opened up to generate concepts on a more latent level. On this basis, it appeared that most of the aforementioned strategies in one way or another influenced the participants' level of arousal. The arousal regulatory aspect of the

training strategy is perhaps most distinct for strategies in the cognitive and emotional regulation domains, where participants aim at raising or lowering mental activity levels in order to steer the object present on the computer screen in the task. However, physiological strategies seemed also to serve this purpose, e.g. decreased activity and muscular activity. It seems that the majority of strategies are used as vehicles to manipulate one's arousal levels, intentionally and/or unintentionally. Hence, the primary mechanism of these strategies is the regulation of the arousal level, which corresponds well with the neurocognitive training objective of NF and rationale of SCP, which is to improve the skills needed to alternate between excitable and inhibitory states.

It is unknown, if the strategies we identified here are specific to SCP-NF or do also apply to other types of NF. It is also unclear to what extent these strategies originate from the attempt to solve the task as such (i.e. the regulation of SCPs) or represent a more general approach to any kind of demand or problem and reactions to feedback. For example, auditory feedback could have generated different strategies than the mostly automated visual feedback. The latter may be particular relevant when investigating NF protocols that offer a greater variety of feedback options (e.g. games, visual/auditory manipulations, continuous/non-continuous feedback, etc.). Future research should investigate strategy use in different NF protocols as well as the effects of different types of feedback on strategy use. Moreover, it is of paramount importance to investigate strategy use in sham-control conditions used in NF trials, as this may help to increase comparability between conditions and to understand the mechanism behind the positive results for some sham-control conditions.

Overall, few trends emerged in our descriptive analysis of the reported strategies over time. However, our results indicate that there are some noteworthy changes. For instance, Breathing was solely reported during the first week, perhaps because this activity is merged into other themes, such as Mindfulness or Focus when the participants emphasize their attention on breathing rather than the activity of breathing, and therefore no longer reported it in this way.

Generally, themes in the physiological domain were moderated via the trainer as most physical activity creates artefacts in the EEG. Before and during SCP training, participants were instructed to avoid movements including facial movements, as these created artefacts in the EEG and disturbed the training procedure. Therefore, Muscular Activity and Breathing might have decreased and Decreased Activity might have increase over time owing to participants following given instructions. Not surprisingly, this pattern was found in the high compliance group, while in the group with lower compliance Muscular Activity was stable over time, and at the booster sessions almost 2/3 acknowledged using it, compared to none among the high compliant subjects. Considering that the trainers discourage muscular activities, it conforms to the level of compliance, as higher compliance included adjustments to such instructions.

Subjects with high compliance were at least four times more likely during the five weeks of SCP training to report Wakefulness strategies compared to subjects with neutral to low compliance. This could be due to higher self-awareness among the highly compliant subjects, and it may also be related to them having a higher overall motivational level, and in effect describe their strategies more elaborately. This may also explain the difference for strategies in the Unspecified domain, as this domain includes many ambiguous statements, and strategies that are more complex and difficult to interpret. As these strategies are more frequently coded for participants with lower compliance, this may be due to their lesser interest and commitment to describe their actions. In addition, using Wakefulness as a strategy may have an inherent connection to the HC group, since the strategy is dependent on a certain level of engagement, as Wakefulness may be assumed costlier in terms of energy expenditure.

We also found noticeable differences between the high compliance group and less compliant participants at the booster sessions. The booster sessions take place shortly before the follow-up assessment for the KITE study, and usually include retrospective comments about the SCP training. Hence, the strategies reported at these sessions are somewhat of a summary of the entire period, and may describe strategies that have been employed previously rather than strategies employed at this specific session. Focus was reported 50% of the time by high compliance subjects, while only 13% of the lower compliance subjects did so. Looking at the domain level, some clear differences emerged. While half of the high compliance subjects reported using Emotional Regulation strategies, only 13% did so among those with lower compliance levels, Emotionally Engaging Thoughts and the use of Specific Emotions were used by 25% compared to none in the lower compliance group. Physiological strategies were much more common among lower levels of compliance compared to subjects with higher compliance.

While our sample was relatively small here, there are some clear trends between the different strategy profiles. The *State of Mind* group had the highest compliance rating, which may have been influenced by age and symptom severity, as the *State of Mind* participants were both older and had a lower Conners 3 scores at start. Higher maturity may have been a factor for increased insight into one's problems and the purpose for the SCP treatment, which in turn would influence once compliance towards the task. Also, it may have been a factor in the ability to relate more abstract to the task and apply strategies that attempt to alter one's mental state.

Interestingly, only participants that were categorized into the *State of Mind* group reported self-perceived improvements, and gave examples of specific situations, mainly within the school context, where they had detected behavioral change in terms of ADHD symptomatology reduction. While for the other two groups such improvements were also reported, they were only observed by their parents or teachers (according to the participant), not noted by the participants themselves. However, it is unclear how well these self-perceived improvements are reflected in symptom reductions on the Conners 3 questionnaires, both short- and long term. Nevertheless, we found some indication of the effects of strategies on relevant behaviors in terms of self-regulation within the SCP training itself [48], with the *State of Mind* profile being the only profile demonstrating a more consistent trend for improved up- and down regulation skills over time, for both the real-time feedback and for transfer trials. Participants with the *Manifest and Concrete* profiles showed the greatest differences between activation and deactivation, and did so already from the first session. This may indicate that these participants are better at self-regulation, however, it may also be the case that the greater differences are obtained through physical manipulations/artefacts, such as the use of Muscular Activity which was common within this profile. This is supported by higher intensity levels being generated in the *Manifest and Concrete* profile strategies compared to the one in the *State of Mind* profile As the current study is exploratory in nature, we are unable at this point to finally determine how strong these observed patterns are and to what extent they overlap with ADHD symptom changes.

Compliance, as evaluated by trainers, based on motivation and participation in the training, appeared to have some influence on strategy use over time. Compliance also differed between the three identified strategy-profiles. Considering the intensity of the SCP-NF in the KITE study (25 sessions in 5 weeks), motivational aspects may have played a key role for training success, even higher than under common, less intense NF settings.

We inquired about the participants' perceived sense of control, which is likely to be closely related to the participants' experiencing of locus of control (LOC). However, the perceived sense of control may also reflect how successful they feel during the NF session. Further

investigations into the relation between LOC and strategy profile is desirable to better understand the association between self-regulation and symptom outcome, as Witte et al. [85] found a negative correlation between LOC scores and the ability to increase sensorimotor rhythm power.

Even though the authors reached consensus regarding the themes and domains, these are still subject to interpretation bias, especially considering the exploratory nature of this study and the limitations encountered by the authors when collecting the data. Firstly, explaining internal, and perhaps at times implicit, mental processes is a priori imprecise, particularly in children with ADHD. Furthermore, a lack of motivation and limited verbal abilities may hinder participants from producing answers that are more elaborate. We observed that participants that were older with higher compliances provided more qualitative material compared to younger participants with lower compliance.

Secondly, one cannot be certain if the description of the themes fully matches the actual activity performed by the subjects. In some interviews, a strategy was described, but it remained uncertain whether the subject actually implemented this strategy, or whether it was just a sensible or wishful description. In other cases, what certain descriptions entailed remained unclear. For instance, in the case of thinking in a direction, it could not be judged whether the participants used a type of imagery (visual or auditory) or whether the employed strategy was more abstract (i.e. an implicit yet intentional activity). Often times we could not be certain whether participants used any of the above, or whether they just provided a trivial answer that reflected the task (e.g. "*I think up, up, up*"). This is especially true for those participants who were younger and with low levels of motivation and language abilities. For them, the interview that occurred directly after a NF session was merely (and sometimes explicitly) an annoying obstacle preventing them from finishing the session and going home.

Differences between some of the themes and sub-themes were small and at times arbitrary. For example, the themes Decreased Activity, Passivity, sluggishness and Mindfulness all describe a state of calmness and inactivity, but are differentiated by their domain (Physical, Cognitive, Emotional or Unspecified). These subtle differences were either directly stated by the participant or provided by the context. Nevertheless, it is possible that some subjects did not differentiate between these variations, and that their statements were ascribed as pertaining to the wrong strategy. The statement "*I do nothing*", for instance, can describe a number of different strategies.

Thirdly, the interaction between trainer and participant is likely to affect their use of strategy, as it is not a silent interaction. Especially, for the participants who are most frustrated over the training will ask what they are supposed to do, hence providing them with examples of what to attempt. However, the analysis of the strategy profiles indicates that the pattern of strategies is rather stable over time.

Fourthly, we do not know whether doing "nothing" (i.e. not applying conscious strategies) is better, as Neumann et al. [55] indicated. Strategies may hinder the learning process as the attention is focused on the correct execution of the strategy rather than on an open-minded evaluation of the strategy's usefulness. This may especially be the case for participants having a limited self-reflective ability, which due to young age would be a large part of the study participants. Furthermore, when asking, "How do you do it?" it is implied that one should implement more explicit strategies.

Finally, the contents of the descriptions of internal mental processes provided by the children with ADHD during the interviews were at times limited. While some participants were thorough in their responses, others did not appear interested in providing elaborated answers. However, due to the exploratory aim of this study this has not been a major issue for our findings, as our goal was to probe and shed light on this neglected area of NF research. Also, as the

interview collection progressed, so did the quality of the questions asked; more follow-up questions were asked and more clarifications given.

We applied a novel until today unique design in research on NF in ADHD. Using repeated short-interviews enabled us to inquire upon internal processes, while at the same time being able to track changes over time. For future research, this could be used complimentary to questionnaires, considering how arduous it can be for many subjects to describe their activities during the SCP training. It therefore seems reasonable to have a researcher/clinician inquiring about these kind of questions verbally, allowing for clarifications. This may especially be useful when combining it with a structured guide and checklist, monitor the use of strategies and changes thereof easier.

Furthermore, we implemented the concept of task-compliance in our study as a broad measure to assess the quality of the task completion/conduction. Initially in the KITE-study, we only asked the subjects to rate their motivation before and after training session, on a scale of 1 to 10. However, the motivation rating was easily influenced by outside factors, such as other activities planned outside of training. In addition, for some younger subjects, and those with more prominent ASD, the concept of motivation was hard to quantify. Hence, we supplemented the motivation self-rating with an overall performance expert rating. Though we believe that our expert rating reflects the subjects' compliance more accurately than a motivation rating, the assessment was blunt, only categorizing compliance trichotomously (poor/fair/ good). For the future, the compliance rating should be elaborated and operationalized with multiple axis, such as; **movement** (in order to assess the generation of muscular artefacts), **motivation** (positive or negative, cares about outcome) and **conformity** (following instruction, adjusting to comments), providing a more precise assessment of how a task is conducted. The latter is important especially in the case of ADHD, where it is most often the parent's motivation and expectations that steer treatment participation, rather than an intrinsic drive by the actual participant. It is therefore of paramount importance for future studies to take into account the level of compliance and style of strategy the participant is prone to use, as this may contribute to predict the outcome.

Our study has shown that there are numerous different strategies applied when performing NF-SCP training. In addition, there seems to be an interaction with the level of compliance, which affects the choice of strategies. Furthermore, there seem to exist different basic styles or profiles of how subjects tackle the NF-SCP task. Six out of 14 reported self-perceived improvements due to the NF-SCP training. This is in line with studies reporting on so-called "performers and non-performers" [86], which to some degree is reflected in the patterns of the manipulation of the slow cortical potentials, at least on a group level.

The exact mechanisms behind NF effects remain unclear. Whether mastering the modification of one's brain activity is the main component, or whether other non-specific factors such as compliance, expectations and the general setting are equal contributors is unknown. In this study, we tried to gain a deeper understand on the NF process via the participant's perspective. By adding the factor of strategy profile, we could get a clearer understanding on the mechanisms behind NF. Some participants, e.g. individuals with the *Manifest/Concrete* profile, may fail regulating their brain activity though improve their symptoms nonetheless. This may be due to the training of sitting still while engaging in a rather boring task, as it is these types of challenges they report when they describe their strategies. Furthermore, some non-responders may not improve their symptoms, as these implemented *State of Mind* strategies, but failed, and therefore did not benefit. Hence, by implementing individual strategy profiles in the evaluation of NF treatments, we might gain further insight into which mechanism are at work at a given moment.

Participants can be seen as naïve learners who via trial-and-error attempt to figure out how to self-regulate their SCPs, which requires a certain level of engagement, or compliance. It is

perhaps therefore that the *State of Mind* profile, with higher levels of compliance, experienced improvements, while the others did not. Nonetheless, it remains unclear whether higher levels of compliance guide the choice of strategy, enables the trial-and-error process, or if the successful implementation of a strategy causes increased compliance. It does raise the question whether we can increase one's compliance by assisting subjects through useful strategies, taking a more coaching approach, or whether we should help the subject find useful strategies themselves, by taking an active approach in working on increasing their compliance and level of self-reflection.

## Supporting information

**S1 Table. List of all domains, themes and sub-themes.** Frequency per theme & sub-theme over time, per session.
(DOCX)

**S2 Table. Summary and coded strategies per session for the *State of Mind* profile.** Session *summaries of each individual interview that is included in the strategy profile.*
(PNG)

**S3 Table. Summary and coded strategies per session for the *Manifest and Concrete* strategy session *summaries of each individual interview that is included in the strategy profile.***
(PNG)

**S4 Table. Summary and coded strategies per session for the *Unaware* strategy profile.** Session *summaries of each individual interview that is included in the strategy profile.*
(PNG)

**S5 Table. Raw data SCP performance.**
(XLSX)

## Acknowledgments

We are grateful for all the participants and their patience during all the interviews. We also gratefully acknowledge Axel D'Angelo, Christer Classon, Johan Hansen Larson, Cecilia Hedin, Soheil Mahdi, Micaela Meregalli, Anna Pilfalk and Shelia Sheikh, for their contributions to the analysis and data collection.

## Author Contributions

**Conceptualization:** John Hasslinger.

**Formal analysis:** John Hasslinger, Manoela D'Agostini Souto.

**Funding acquisition:** Sven Bölte.

**Investigation:** John Hasslinger, Manoela D'Agostini Souto, Lisa Folkesson Hellstadius.

**Methodology:** John Hasslinger.

**Project administration:** John Hasslinger.

**Supervision:** Sven Bölte.

**Validation:** John Hasslinger, Manoela D'Agostini Souto, Lisa Folkesson Hellstadius.

**Writing – original draft:** John Hasslinger, Manoela D'Agostini Souto, Lisa Folkesson Hellstadius.

**Writing – review & editing:** Sven Bölte.

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
