## [Decision Letter · Decision Letter 0]

10 Feb 2020

PONE-D-19-34387

Neurofeedback in ADHD: a qualitative study of strategy use in slow cortical potential training

PLOS ONE

Dear Mr. Hasslinger,

Thank you for submitting your manuscript to PLOS ONE. After careful consideration, we feel that it has merit but does not fully meet PLOS ONE’s publication criteria as it currently stands. Therefore, we invite you to submit a revised version of the manuscript that addresses the points raised during the review process.

The three reviewers make some valid points regarding framing and report of the results. Two of them also ask for additional report regarding the link between regulation strategies and training outcome (objective rather than subjective). I suggest that you consider those additional reports.

We would appreciate receiving your revised manuscript by Mar 26 2020 11:59PM. To enhance the reproducibility of your results, we recommend that if applicable you deposit your laboratory protocols in protocols.io, where a protocol can be assigned its own identifier (DOI) such that it can be cited independently in the future. For instructions see: http://journals.plos.org/plosone/s/submission-guidelines#loc-laboratory-protocols

We look forward to receiving your revised manuscript.

Kind regards,

Hedwig Eisenbarth

Academic Editor

PLOS ONE

Journal Requirements:

Reviewers' comments:

Reviewer's Responses to Questions

**Comments to the Author**

1. Is the manuscript technically sound, and do the data support the conclusions?

Reviewer #1: Yes

Reviewer #2: Partly

Reviewer #3: Partly

2. Has the statistical analysis been performed appropriately and rigorously? 

Reviewer #1: N/A

Reviewer #2: N/A

Reviewer #3: N/A

3. Have the authors made all data underlying the findings in their manuscript fully available?

Reviewer #1: No

Reviewer #2: No

Reviewer #3: No

4. Is the manuscript presented in an intelligible fashion and written in standard English?

Reviewer #1: Yes

Reviewer #2: Yes

Reviewer #3: Yes

5. Review Comments to the Author

Reviewer #1: The authors present a very interesting qualitative study into strategy use during SCP neurofeedback in ADHD. I agree that these kind of studies are important and long overdue, and I am hopeful that these insights will contribute to a better and more nuanced understanding of neurofeedback as treatment for ADHD. The manuscript is well written and the methods seem sound, I mostly have minor comments and ideas for additional discussion.

Introduction

1. In the explanation of neurofeedback (lines 75-77), you may add that other important ‘ingredients’ are learning principles, such as operant conditioning.

2. The first sentence of the paragraph (starting at line 90) does not really represent the rest of the paragraph, which mostly sums up the many limitations in neurofeedback research.

3. When discussing specific versus non-specific effects, which is at the core of the neurofeedback efficacy ‘debate’, you may want to consult the CRED-nf checklist. Recently, a consensus report was drafted on the reporting and experimental design of neurofeedback studies. The report contains a useful model that distinguishes multiple mechanisms that together may drive the total effect of neurofeedback, including (1) neurofeedback specific, (2) neurofeedback non-specific, (3) general non-specific, (4) repetition-related and (5) natural mechanisms.

Methods

1. Although the authors provide more details on the specific instructions that participants received during the neurofeedback session in the discussion, please also provide those in the methods section (were there any instructions in the protocol?): (a) How was the SCP training framed? As a way to treat ADHD, to self-regulate attention?, (b) How were the participants instructed concerning physiological artefacts?, (c) What instructions did they receive during the sessions?

2. On lines 241-242, it is described that mostly successful strategies were explored. Although this makes sense, do the authors think future research should also ask about what strategies did not work, and how many strategies were used during each session (as a measure of trial-and-error learning)?

Results

1. I think most results are presented well, but there is some inconsistent use of the words ‘theme’ and ‘sub-theme’ throughout the results section, and Table 2 (which ones are italic for example), making it more difficult to follow.

2. It may be useful to provide a ‘overview’ figure with all domains, and the themes and subthemes (this could be depicted using hierarchical boxes on top of each other).

3. Line 373: also name the number of themes and subthemes, including their names (like the other domain paragraphs).

4. I believe the subtheme ‘silence’ is not discussed in the text (of physiological strategies), while ‘breathing’ twice.

Discussion

1. It is fascinating to recognize many strategies, as reported in this manuscript, in my own experience with theta/beta neurofeedback research in ADHD and conversations I had with participants. Although SCP and theta/beta neurofeedback have different targets, both seem to affect (or regulate, in SCP) arousal. It may be interesting to discuss what the results mean for other neurofeedback protocols.

2. The authors discuss three different strategy ‘profiles’. The ‘state of mind’ profile is associated with intrinsic motivation, which may be an important factor in the ultimate efficacy of the training (although this is not studied in the current manuscript). It would be very interesting if the authors discuss the role of motivation more extensively. For example, the role of self-efficacy, locus-of-control, etc. in ADHD and as potential moderator for strategy use and efficacy of neurofeedback in ADHD.

3. Would you recommend exploring strategy use in EMG biofeedback (as sham-control condition for neurofeedback)? It may be interesting to see whether there is an overlap in strategies used, which may explain non-specific effects of neurofeedback.

4. Line 729: typo (the word ‘not’ is missing between ‘did’ and ‘benefit’).

Tieme Janssen

Reviewer #2: The manuscript entitled „Neurofeedback in ADHD: a qualitative study of strategy use in slow cortical potential training“ reports on the results of a qualitative explorative investigation of strategies during SCP-NF in 30 children and adolescents with ADHD. The authors derived strategy clusters covering cognitive, physiological, emotional and unspecified domains. Furthermore, from 14 participants, three strategy profiles could be differentiated: “State of Mind”, “Manifest and Concrete” and “Unaware”. This is an interesting study with the rationale to shed light on NF mechanisms. The manuscript is well written and clearly structured, and I believe that the results are of interest to other researchers in the field.

However, it would seem important to link the qualitative data (/subjective experience) at least to SCP regulation performance (i.e. amplitudes as an objective measure). The mere description of strategies doesn’t help evaluating how efficacious they are on NF performance. Thus, the final conclusion that the findings may help to facilitate more effective instructions in how to approach SCP is not supported by the data. From the data provided in the current version of the manuscript, we can’t see which strategies avail or hinder treatment effects or if using explicit strategies would be useful at all. Thus, we learn from the manuscript which strategies are used, but we do not know which of them are advantageous.

I believe that it would increase the informative value for the scientific community tremendously if the authors would build the bridge between the strategies or the strategy profiles and NF performance (cf., e.g., Kober et al., 2013) as well as symptom changes. This is, in my opinion, the most important part as it gives information on:

- Which strategies/ strategy profiles are (not) effective ((not) leading to negative or positive SCP shifts)? Can strategies predict outcome (Conners)?

- Are there differences in which strategies are particularly effective/feasible in everyday life (leading to more pronounced changes in Conners)?

o e.g., negative emotional strategies (anger, sadness) might be less feasible in everyday life as inducing negative emotions to take yourself into a state of attention might not be the appropriate approach

- Are there differences in which strategies are particularly effective depending on presentations of ADHD (Conners subscales)?

o e.g., the physiological strategies most probably constitute a general physical arousal regulation rather than regulation of a circumscribed disorder-specific brain region (thereby leading to specific ADHD-related symptom changes). Still, they might be particularly effective in reducing motor hyperactivity (by general relaxation).

- Is lack of insight an advantage (leading to more pronounced amplitudes and speaking in favor of an automatic learning process)?

Minor:

Reports from teachers or parents occurred by report of the children?

How did these reports coincide with the ratings by teachers or parents themselves?

In Figure 1 SCP Feedback. Looking at the feedback screen depicted in the middle, it looks as if a threshold would have been implemented during training, which is not mentioned in the manuscript and supposedly was not the case.

The authors should recheck spaces (e.g. line 50, 56, 94, 254) and misplaced words (e.g., line 152, 603).

In line 530, it should read “(…) frequently, showing that many strategies (…)”

Reviewer #3: In this manuscript, the authors analyzed quantitatively [editor: I think they meant qualitatively] the self-regulation strategies during specific and standardized neurofeedback training in ADHD patients. This is an important study which adds relevant and highly novel insights to the field of neurofeedback training. The main aims were to map the cognitive strategies, evaluate the changes over time and to investigate different profiles.

The manuscript is well written, but the result section is difficult to follow. In my opinion, it would be helpful to include some tables in which the proportion of subjects and the different strategies are stated and classified.

One major concern needs to be addressed to be suitable for publication and therefore to add novel insights to the field. In my opinion, the authors must include the factual self-regulation data of the NF training and performance. This is important to be able to answer the question of which cognitive or emotional strategy worked. Now the manuscript only is able to show in a quantitatively [editor: I think they mean qualitatively here] way that the participants tried something, but the key question remains unanswered! Did a specific strategy work? Did one work better? – In my opinion, this is really important and would lead to a very important paper. The authors might just look at the mean % of suns delivered by Theraprax?

If the self-regulation data is not included, the authors should address this issue in more detail during the discussion or even add a limitation section.

Minor:

Introduction

L70 Citation error

L81 ADHD symptoms as well as for medical conditions the authors may add a citation

L93 – The authors should cite more recent meta-analysis (ie. Cortese et al., 2016, Riesco-Mejias et al 2019)

L128 The authors should consider as well following manuscript when addressing the Placebo question in NF (Busalb et al 2019 https://www.frontiersin.org/articles/10.3389/fpsyt.2019.00035/full)

L131 The authors should specify that this study was done in Adults

Methods:

Did the participants receive some instruction prior to the SCP-NF training?

I miss the differentiation between strategies for the different conditions. SCP-NF has two conditions which are Negativity and Positivity trials. Did the authors look if there were different strategies during those different conditions? - What about the transfer vs. real-time feedback?

Discussion:

L 629 The authors state that they do not know if the subjects succeed in modulating their brain activity. This is a critical point and in my point of view if included it will enrich the manuscript and the conclusions.

L634 “While the other two groups also improvements were reported” I think that here is missing something.

L717 though we do not know whether these individuals were successful in manipulating their slow cortical potentials. The authors must analyse that.

6. PLOS authors have the option to publish the peer review history of their article (what does this mean?). If published, this will include your full peer review and any attached files.

Reviewer #1: Yes: Tieme Janssen

Reviewer #2: No

Reviewer #3: No

---

## [Author Response · Author response to Decision Letter 0]

6 Apr 2020

Reviewer #1

Thank you very much, we truly appreciate your recognition of the importance of our research.

Introduction 1: Thank you, other learning principles have been added to the introduction.

Introduction 2: Thank you for addressing this, we have rewritten the opening sentence for clarity.

Introduction 3: Thank you very much for bringing this to our attention. We have completed the section where we discussed specific versus non-specific effects, to also mention repetition-related and natural mechanisms. We now also cite the CRED-nf checklist here, which is an excellent reference in the given context.

Methods 1: Thank you for this very valuable request for clarification regarding the points a, b, and c. We have elaborated the “SCP task” section, and added information addressing the framing of the training and the instructions that were given to the participants. 

Methods 2: Thank you and apologies for the lack of clarity. Non-effective strategies were also addressed by some participants, when we followed up the participants’ answers with the standard question: “how did that work for you”. Thus, both successful and less successful strategies experienced by the participants were assessed. Nonetheless, a more focused, elaborated and explicit inquiry about the non-successful strategies would surely have been desirable and should be included in future research. The number of strategies reported in each interview is shown in the supporting information Tables S2-S4.

Results 1: Thank you for pointing this out to us. There were some inconsistencies in using italics that have been corrected. In addition, we added a legend to table 2 and 4 to further clarify the themes and subthemes.

Results 2: We agree that a figure showing the pattern of themes and subthemes may be useful, and have added a figure illustrating the theme structure early in the result section. 

Results 3: Done.

Results 4: True - thank you for noticing. Silence was a “isolated” subtheme and only defined by one quote. It was removed due to its limited significance, and its defining statement was later moved to “Passivity” as the essence was to be quiet. We had removed it from the text but had missed removing it from the table, which now has been done.

Discussion 1: Very interesting to hear that the reviewer shares our experience regarding the strategies for T/B NF. In the KITE-study, we have also gathered qualitative data on strategies using interviews for live z-score NF. Initially, we had planned to compare them with SCP in one paper, but we had to cancel that plan since the amount of information would have been to extensive to wrap in one journal manuscript. Instead the strategies used in z-score training will be presented in a separate later paper. Still, in order to address the reviewer’s valid point, we have added some lines on this topic to the paper’s discussion sections. 

Discussion 2: We agree completely, that these are very relevant suggestions, why we have extended the discussion of motivation as recommended in the respective section. Furthermore, we will address these questions more thoroughly in a future paper that is in progress.

Discussion 3: Thank you for your input. We agree that it would be especially interesting to explore strategies in sham-control conditions, considering that they often also lead to symptom improvements. Accordingly, this aspect has been added to the discussion section. 

Discussion 4: Corrected, thank you, very attentive of you.

Reviewer #2

Thank you for all the positive and encouraging comments. 

Yes, this is a very important point you make, and we have added an exploratory analysis of the SCP regulation performance for the three types of strategy profiles. A further analysis of the impact of individual strategies on regulatory performance would be very interesting for future studies, with the current data-set, however, it is not possible. 

We agree that linking the different elements of 1) strategy use, 2) NF performance (self-regulation) and 3) symptom outcomes is highly significant. Please notice, however, that the nature of this study is qualitative (!) and exploratory, more generating hypotheses, than testing any with regard to strategy use. The data that the reviewer is requesting would rather be meaningful to be collected in an follow-up study based on the current findings. Nonetheless, we have now added data to the paper linking strategies and performance as much as possible and reasonable. We hope the reviewer finds it informative that we provide self-regulation performance in SCP for the three derived strategy profiles. However, this information is also of rather exploratory and preliminary nature. 

Minor 1: Yes, all statements are based on the interviews with the children and adolescents. 

Minor 2: Correct, with few exceptions there was a threshold implemented during the training. We have clarified this in the section addressing the “SCP task”.

Minor 3: Thank you, very attentive of you. It has been corrected.

Reviewer #3

We are grateful for your recognition of our study.

We apologize for the lack of clarity. Table S1 in the supporting information lists all strategies (themes and sub-themes) including the frequency per session. We had this table in the main text in earlier drafts of the manuscript. We have also added an overview figure, which we hope will help.

Please see also our more extensive reply to reviewer #2.

Thank you – the reviewer raises of course relevant questions. To comply with these we have added information on the SCP regulation performance for the different derived strategy profiles. Nonetheless, please note that this study is a qualitative one, of exploratory nature and personal experiencing, rather generating hypotheses of used strategies, than testing the effects of established strategies.

Minor

Introduction 1: Thank you, this has been corrected.

Introduction 2: Thank you, the sentence has been revised.

Introduction 3: Thank you very much for bringing this to our attention. We have included the Riesco-Matías et al meta-analysis.

Introduction 4: Thank you for bringing the Bussalb et al article to our attention. We have incorporated it in the text.

Introduction 5: Thank you, we have clarified this.

Discussion 1: Agreed; this is indeed highly relevant. Though we pursue an exploratory, qualitative approach, we have added a brief examination of the self-regulation performance during SCP for the three derived strategy profile groups.

Discussion 2: Thank you for pointing this out. We have elaborated the instructions given to the participants in the method.

Discussion 3: Yes, this was something that we addressed in our initial analysis phase, when the verbal material was structured based on the conditions. Still we merged the material again subsequently, as many strategies, for both Negativity and Positivity, are simply inverted versions dependent on the direction. 

In addition, as for transfer/real-time feedback, the extent and depth of the responses given by the child and adolescent participants was not rich enough to go into such detail in a meaningful way. Instead, we focused on broader themes that are describing strategies used for all conditions.

Discussion 4: Yes, it should have read: “While the other two groups also reported improvements,…”. This has been corrected.

Discussion 5: See also our responses above. We hope that the newly added data on the SCP regulation addresses this issue sufficiently. Again, this is predominately a qualitative study.

---

## [Decision Letter · Decision Letter 1]

5 May 2020

Neurofeedback in ADHD: a qualitative study of strategy use in slow cortical potential training

PONE-D-19-34387R1

Dear Dr. Hasslinger,

We are pleased to inform you that your manuscript has been judged scientifically suitable for publication and will be formally accepted for publication once it complies with all outstanding technical requirements.

With kind regards,

Hedwig Eisenbarth

Academic Editor

PLOS ONE

Additional Editor Comments (optional):

Reviewers' comments:

Reviewer's Responses to Questions

**Comments to the Author**

1. If the authors have adequately addressed your comments raised in a previous round of review and you feel that this manuscript is now acceptable for publication, you may indicate that here to bypass the “Comments to the Author” section, enter your conflict of interest statement in the “Confidential to Editor” section, and submit your "Accept" recommendation.

Reviewer #2: All comments have been addressed

Reviewer #3: All comments have been addressed

2. Is the manuscript technically sound, and do the data support the conclusions?

Reviewer #2: Yes

Reviewer #3: Partly

3. Has the statistical analysis been performed appropriately and rigorously? 

Reviewer #2: N/A

Reviewer #3: Yes

4. Have the authors made all data underlying the findings in their manuscript fully available?

Reviewer #2: No

Reviewer #3: Yes

5. Is the manuscript presented in an intelligible fashion and written in standard English?

Reviewer #2: Yes

Reviewer #3: Yes

6. Review Comments to the Author

Reviewer #2: The authors addressed my comments adequately thus I would recommend publication of the MS in its current form in PLOS ONE.

Reviewer #3: The authors of this MS have been very responsive to my comments and have submitted an excellent revision.

7. PLOS authors have the option to publish the peer review history of their article (what does this mean?). If published, this will include your full peer review and any attached files.

Reviewer #2: No

Reviewer #3: No

---

## [Editor Report · Acceptance letter]

26 May 2020

PONE-D-19-34387R1 

Neurofeedback in ADHD: a qualitative study of strategy use in slow cortical potential training 

Dear Dr. Hasslinger:

I am pleased to inform you that your manuscript has been deemed suitable for publication in PLOS ONE. Congratulations! Your manuscript is now with our production department. 

With kind regards,

on behalf of

Dr. Hedwig Eisenbarth 

Academic Editor

PLOS ONE